# ATO Increases ROS Production and Apoptosis of Cells by Enhancing Calpain-Mediated Degradation of the Cancer Survival Protein TG2

**DOI:** 10.3390/ijms241310938

**Published:** 2023-06-30

**Authors:** Károly Jambrovics, Szilárd Póliska, Beáta Scholtz, Iván P. Uray, Zoltán Balajthy

**Affiliations:** 1Department of Biochemistry and Molecular Biology, Faculty of Medicine, University of Debrecen, Egyetem tér 1, H-4032 Debrecen, Hungary; jambrovics.karoly@med.unideb.hu (K.J.); uray.ivan@med.unideb.hu (I.P.U.); 2Genomic Medicine and Bioinformatic Core Facility, Department of Biochemistry and Molecular Biology, Faculty of Medicine, University of Debrecen, H-4032 Debrecen, Hungary; poliska@med.unideb.hu (S.P.); scholtz@med.unideb.hu (B.S.); 3Department of Clinical Oncology, Faculty of Medicine, University of Debrecen, Egyetem tér 1, H-4032 Debrecen, Hungary

**Keywords:** ATO, APL, cancer, cell survival, cell death, reactive oxygen species, TG2

## Abstract

Transglutaminase 2 (TG2) is a critical cancer cell survival factor that activates several signalling pathways to foster drug resistance, cancer stem cell survival, metastasis, inflammation, epithelial-mesenchymal transition, and angiogenesis. All-trans retinoic acid (ATRA) and chemotherapy have been the standard treatments for acute promyelocytic leukaemia (APL), but clinical studies have shown that arsenic trioxide (ATO), alone or in combination with ATRA, can improve outcomes. ATO exerts cytotoxic effects in a variety of ways by inducing oxidative stress, genotoxicity, altered signal transduction, and/or epigenetic modification. In the present study, we showed that ATO increased ROS production and apoptosis ratios in ATRA-differentiated NB4 leukaemia cells, and that these responses were enhanced when TG2 was deleted. The combined ATRA + ATO treatment also increased the amount of nuclear factor erythroid 2-related factor 2 (NRF2) transcription factor, an adaptive regulator of the cellular oxidative stress response, and calpain proteolytic activity, resulting in TG2 degradation and the reduced survival of WT leukaemia cells. We further showed that the induced TG2 protein expression was degraded in the MCF-7 epithelial cell line and primary peripheral blood mononuclear cells upon ATO treatment, thereby sensitising these cell types to apoptotic signals.

## 1. Introduction

Transglutaminase 2 (TG2) is a multifunctional enzyme with GTPase, ATPase, protein kinase, protein disulfide isomerise, and transglutaminase activities. It is expressed in several cell types and organelles, carrying out various posttranslational modifications, such as protein crosslinking, amine incorporation, acylation, and deamidation [1]. TG2 can also serve as a scaffold protein that connects protooncogenes, the tumour suppressor PTEN, integrins, and growth factor receptors intracellularly and extracellularly [2]. TG2 is involved in a number of cellular processes, such as differentiation, apoptosis, phagocytosis, signal transduction, adhesion, and wound healing [3]. Growing evidence now indicates an association between increased TG2 expression and several human cancers in which TG2 can function as a fundamental cancer cell survival factor. In patients with lung, breast, colon, cervical, leukaemia, lymphoma, and pancreatic cancers, an elevated expression of TG2 is associated with poor prognosis, metastasis, and disease recurrence [4,5,6,7,8].

Acute promyelocytic leukaemia (APL) is a bone marrow cancer that is characterised by the presence of a chromosomal translocation between the retinoic acid receptor alpha (RARα) gene on chromosome 17 and the promyelocytic leukaemia protein (PML) gene on chromosome 15. The PML-RARα oncoprotein acts as a transcription corepressor by blocking myeloid cell differentiation. The treatment of APL cells with all-trans retinoic acid (ATRA) triggers a conformational change in the oncoprotein and the subsequent proteolysis of the PML-RARα fusion protein, thereby shifting leukemic cells from the transcriptionally repressed state to the activated state and resulting in their maturation, while several genes are upregulated [9,10,11]. To some extent, ATRA-induced APL differentiation can be modelled using NB4 WT APL cells. To produce functioning neutrophil granulocytes, hundreds of genes are modulated during the differentiation process, among which TG2 is the most strongly upregulated gene in ATRA-activated NB4 cell maturation [9,10,11,12,13].

In the 2000s, the US Food and Drug Administration (FDA) authorised arsenic trioxide (As_2_O_3_/ATO) as a treatment for APL. Several clinical studies have shown that ATO, either as a single agent or combined with ATRA, generates an exceptionally better result than ATRA alone [14,15,16], making ATO a first-choice option for the treatment of newly diagnosed and relapsed APL patients. Arsenic treatment can induce complete remission in patients and provide a 5-year overall survival rate of 90% when it is combined with ATRA and chemotherapy [17]. In addition, ATO is an outstanding monotherapy, as it eliminates residual leukaemia-initiating cells (LICs), a result that cannot be achieved by ATRA treatment alone. The cytotoxic effects of ATO are concentration-dependent; at low concentrations (<0.5 μM), ATO can induce the partial differentiation of APL cells, and at high concentrations (>0.5 μM) it initiates apoptosis. At the cellular and molecular levels, ATO can provoke the degradation of the PML-RARα fusion protein by targeting the PML part and inducing sumoylation, ubiquitination, and the proteasomal degradation of the chimeric protein [18,19,20]. In addition, ATO generates both oxidative stress and damage to cellular biomolecules, including DNA, lipids, and proteins, to cause cell death [20]. Ample evidence also indicates that some transcription factors and signal transduction pathways are redox-sensitised by arsenic [21]. Oxidative stress physiologically triggers the cell’s antioxidant responses by activating transcription factors, e.g., NRF2. NRF2’s translocation into the nucleus and later gene expression regulation activates antioxidant genes [22,23].

One group of proteins that might be sensitive to ATO is the calpain family, whose members are ubiquitously expressed in almost all eukaryotes. Calpains are Ca^2+^-dependent cysteine proteases that display proteolytic activity and hydrolyze various substrate proteins as membranes or in the cytosol. Although the physiological role of calpains is still poorly understood, they are active participants in cellular processes such as cell apoptosis [24]. Here, we report that ATO-induced ROS induced both the transcription factor NRF2 (nuclear factor erythroid 2-related factor 2) protein and the proteolytic enzyme calpain, whose activation ultimately leads to the proteolysis of TG2.

## 2. Results

### 2.1. Transglutaminase 2 Impedes ROS Production to Attenuate ATO-Induced Cell Death in ATRA-Differentiated NB4 Cells

ATO induces the production of reactive oxygen species (ROS) as one of its anti-tumour effects. The ROS levels in NB4 WT cells gradually decreased after the first day of the ATRA and ATRA + ATO treatments, but they increased following ATO treatment alone (Figure 1A). Similar endogenous ROS production patterns were seen in NB4 TG2-KO cells upon ATO treatment alone (Figure 1B).

Increased ROS production was associated with an elevated apoptosis ratio. In NB4 WT cells, TG2 expression was induced by ATRA and was associated with a lower rate of cell death (Figure 2A, orange and grey bars compared to blue bars) [12]. By contrast, ATO resulted in higher apoptosis rates in the absence of TG2 induction in NB4 TG2-KO cells compared to in NB4 cells (Figure 2B).

### 2.2. ROS-Induced Nuclear Factor Erythroid 2-Related Factor 2 (NRF2) Transcription Factor Mediates the Expression of Antioxidant Genes in a TG2-Dependent Manner

The NRF2 transcription factor plays a crucial role in the adaptive cellular oxidative stress response by strengthening the antioxidant and detoxification systems. The ATRA + ATO treatment increased NRF2 mRNA and protein expression in NB4 WT cells, but this induction was blunted in TG2 KO cells (Figure 3A,B). Glutathione S-transferase omega-1 (GSTO1-1) plays an essential role in the arsenic biotransformation of the arsenic methylation pathway. Peroxiredoxin 5 (PRDX5) is an antioxidant enzyme with a cytoprotective function during oxidative stress. GSTO1 mRNA was accumulated equally following all treatments in NB4 WT cells, but not in the absence of TG2 (Figure 3C,D). While the amount of GSTO1 protein in ATRA-treated cells did not change significantly from day 2, the combined treatment of ATRA + ATO gradually increased the GSTO1 protein levels (Figure 3C). The only treatment that caused a change in the PRDX5 mRNA levels was the combination of ATRA + ATO, and the induction was abolished in the absence of TG2, just as it was with GSTO1 (Figure 3D,F). The protein levels of PRDX5 correlated with the transcript levels (Figure 3E,F).

### 2.3. ROS Production Elicits Robust Calpain Expression and Activation, Resulting in the Increased Proteolysis of Transglutaminase 2

Calpains are ubiquitous non-lysosomal calcium-dependent cysteine proteases activated during various pathological and physiological processes (e.g., apoptosis and cell proliferation). ATRA treatment was associated with constant calpain expression, but the ATRA + ATO treatment markedly increased the levels of calpain starting from day 2 and showed ATO concentration dependence (Figure 4A).

Fluorimetry determinations showed increased calpain enzymatic activity following ATRA or ATRA + ATO treatment in both the total cell population (Figure 4B, grey bars) and after sorting the Annexin-V-negative (green) and Annexin-V-positive (pink) cells. By day three, calpain activity in the total cell population was 6.28-fold higher in the ATRA + ATO-treated cells than in the ATRA-treated cells. When cells were separated by apoptotic features, the calpain activity was 3.25-fold higher by day 3 and 24.4-fold higher by day 5 in Annexin-V–positive cells than in Annexin-V–negative cells (Figure 4B). Calpain protein expressions followed the measured activity increases in Annexin-V-positive cells (Figure 4C).

The treatment of NB4 WT cells with ATRA or ATRA + ATO in the presence of 1 and 2 μM calpain inhibitor for five days resulted in only a minimal fraction of ATRA-treated cells showing signs of apoptosis, whereas the effect of the ATRA + ATO treatment, which induced cell death in close to 60% of the cells, was reduced by about 15 and 20% by the presence of 1 and 2 μM calpain inhibitor, respectively (Figure 4D). In parallel, treatment with a 2 μM calpain inhibitor reversed the ATO-induced calpain-mediated TG2 degradation, as indicated by increases in TG2 to levels comparable to those seen with ATRA alone (Figure 4E).

### 2.4. ATO Could Suppress Both Transglutaminase 2 mRNA and Protein Levels in NB4 WT, MCF-7 Cells, and in MCSF-Treated Monocytes

The NB4 WT cell line was differentiated with ATRA with or without ATO at 0.5 or 2.0 μM for 5 days. ATRA alone induced a continuously increasing TG2 expression, which was constantly mitigated at both the mRNA and protein levels by the presence of ATO. The amount of TG2 protein showed a dose-dependent, but not statistically significant, increasing trend (Figure 5A,B).

We tested whether this response was cell line specific by examining doxycycline-induced TG2 expression in terms of mRNA and protein expression in the MCF-7 breast carcinoma cell line. Compared to the ATRA-treated samples, the ATRA + ATO treatment reduced TG2 mRNA expression by more than 50% and protein expression by more than 75%. As a positive control for TG2 expression, the NB4 WT cells showed the same reduction in TG2 expression as was observed in MCF-7 breast carcinoma cells (Figure 5C, blue and black bars, Figure 5D). Finally, human monocytes were isolated from three healthy donors and treated for 5 days with MCSF, ATO, calpain inhibitor, or combinations ex vivo to validate the ATO-induced TG2 degradation seen in the commercial cell lines. Upon MCSF treatment, the monocytes expressed gradually increasing amounts of TG2 mRNA on days 3 and 5, and this expression was significantly decreased by ATO-induced calpain activation (Figure 5E). The use of the calpain inhibitor preserved the TG2 protein levels at days 3 and 5 (Figure 5F). TG2 protein expression showed a similar pattern to that of TG2 mRNA, but the ATO-induced TG2 degradation was more moderate and was completely inhibited by the calpain inhibitor treatment. Upon arsenic-induced oxidative stress in NB4 WT cells, the degradation of RNA was more significant, but the degradation dynamics were different (Appendix A). In samples treated with both MCSF and a calpain inhibitor, the inhibitor stopped the natural breakdown of TG2, resulting in the highest TG2 expression (Figure 5F,G).

## 3. Discussion

The American Cancer Society estimated that 1.9 million new cancer cases would be diagnosed in the United States in 2022, and 609,360 cancer deaths would occur. Despite significant advances in cancer treatment in recent years, tumour recurrence, drug resistance, and metastasis remain significant issues. Consequently, new treatment options are still desperately required.

For individuals with acute promyelocytic leukaemia, a combination of arsenic trioxide and all-trans retinoic acid has been demonstrated to be an effective therapy choice. ATRA is a drug that can stimulate the differentiation of promyelocytes, whereas ATO can trigger programmed cell death in APL cells, particularly those with the PML-RARalpha fusion gene [25]. When applied to patients with APL illness, the combination of ATRA and ATO therapy has the potential to enhance overall survival rates in addition to increasing the likelihood of reaching complete remission [26]. This combination medication has been shown to be successful in a number of clinical trials, particularly in individuals who have just been diagnosed with APL [27].

Our findings showed that TG2 expression was inversely correlated with the elevated apoptosis ratio induced by increased ROS production in response to ATRA + ATO treatment (Figure 1A,B, Figure 2A,B and Figure 3A,B). The standard first-line treatment for APL has been a combination of all-trans retinoic acid (ATRA) plus chemotherapy. ATO exerts its cytotoxic effects in several ways, including oxidative stress, genotoxicity, the alteration of the signal transduction cascade, and epigenetic modification. In addition, TG2 promotes the expression of genes responsible for the elimination of reactive oxygen and nitrogen species through increased transcription and translation of the NRF2 transcription factor that coordinates the stress-inducible activation of cytoprotective genes [28]. Consequently, the absence of TG2 expression greatly increases ATO-induced oxidative stress (Figure 3A–F).

Here, we demonstrated that the combination of ATRA and ATO, but not ATRA treatment alone, could increase the amount and proteolytic activity of calpain, thereby driving both the degradation of TG2 and the decreased survival of leukemic cells (Figure 4A–E). Finally, we showed that the degradation of TG2 is not cell-lineage specific by showing that induced TG2 protein expression can be broken down in myeloid-derived ATRA-differentiated NB4 WT cells in the MCF-7 breast cancer cell line and primary peripheral blood mononuclear cells, thereby sensitising each of these cell types to apoptotic signals (Figure 5A–G).

Analysing the expression of TG2 in nearly 3000 solid tumours led to the conclusion that in tumours with elevated TG2 levels, the prognosis was worse in terms of overall survival (OS); disease-free (DFS) and recurrence-free survival (RFS) was shorter, with an almost double hazard ratio (HR). These findings suggest that ATO treatment might improve the prognosis and decrease the likelihood of metastasis and recurrence in a variety of cancers after surgery [6,18,29,30,31].

Finally, these data could imply that ATO treatment might improve the prognosis and decrease the likelihood of cancer metastasis in a variety of cancers, and so treatment with arsenic trioxide could be a new option for tumour therapies with atypical TG2 expression, including colon, breast, stomach, leukaemia, lymphoma, oesophagus, pancreatic, pulmonary, and cervical cancers [32].

## 4. Materials and Methods

### 4.1. Cell Culture and Treatment

NB4 cell lines were cultured in RPMI 1640 (HEPES-containing) medium (Sigma-Aldrich, St. Louis, MO, USA), while MCF-7 cells were cultured in Dulbecco’s modified Eagle’s medium (DMEM; Sigma-Aldrich) supplemented with 10% (*v*/*v*) foetal bovine serum (Gibco, Paisley, Scotland), 2 mM L-glutamine, 1% (*v*/*v*) 100 U/mL penicillin-streptomycin, and 1% sodium pyruvate solution (10 mg/mL) (Sigma-Aldrich) at 37 °C in 5% CO_2_. All cell lines were periodically tested for mycoplasma contamination.

The NB4 cells were treated with 1 μM all-trans retinoic acid (ATRA) (Sigma-Aldrich) alone or in combination with 0.5 μM or 2.0 μM arsenic trioxide (ATO) for 5 days.

MCF-7 cells were treated with 1 µg/mL doxorubicin (Sigma Aldrich), 1 μM ATRA, 1 μM ATRA + 1 µg/mL doxorubicin, or 1 µg/mL doxorubicin + 2.0 μM ATO for 3 and 5 days.

Human monocytes were treated with 2.0 μM ATO, 2.0 μM calpain inhibitor, 5 nM macrophage colony-stimulating factor (MCSF), 5 nM MCSF + 2.0 μM ATO, 5 nM MCSF + 2.0 μM ATO + 2.0 μM calpain inhibitor, or 5 nM MCSF + 2.0 μM calpain inhibitor for 3 and 5 days.

### 4.2. Human Monocyte Isolation

Ficoll-gradient centrifugation was used to separate CD14+ human monocytes from healthy donors, and then anti-CD14-conjugated microbeads were used to separate immune cells. The supernatant was discarded after blood samples were spun at 700× *g* for 15 min at room temperature. The remaining samples were diluted two-fold with physiological saline (Sigma Aldrich). The diluted samples were layered on 10 mL Ficoll reagent and centrifuged at 700× *g* for 20 min at room temperature. The resulting Ficoll layer containing monocytes (approximately 30 mL) was transferred into a new tube, diluted with 20 mL physiological saline, and centrifuged for 10 min at room temperature at 350× *g*. After centrifugation, the pellet was diluted, resuspended in 9 mL physiological saline, and filtered through a column prefilter. The filtered cells were centrifuged for 7 min at room temperature at 350× *g*. Next, the pellet was resuspended in 800 mL of 1 × phosphate-buffered saline (PBS) supplemented with 200 μL CD14+ microbeads (R&D). The cells were incubated for 20 min at 4 °C. The labelled cells were removed by magnetic separation columns provided by R&D, followed by centrifugation for 7 min at 700× *g*. The pelleted cells were resuspended, counted, and plated for further experiments. Monocytes were cultured in multi-well culture plates in Iscove’s modified Dulbecco’s medium (IMDM) supplemented with 10% human AB serum (Invitrogen). For macrophage differentiation, freshly plated monocytes were treated with 5 nM MCSF (0.5 μL/mL). Cells were harvested at 0, 3, and 5 h after treatment.

### 4.3. Inhibition of Calpain

Untreated, ATRA-differentiated, and ATRA + 2.0 μM ATO-treated NB4 wild-type (WT) cells were treated with 1 μM or 2 μM calpain inhibitor (Bio-Rad, Hercules, CA, USA) for 3 and 5 days.

### 4.4. Endogenous Reactive Superoxide Production

A total of 200 μL reaction mixture containing 1 × 10^5^ cells and 2 μL dichloro-fluorescein diacetate (DHCFDA) (200 μM) were combined and incubated for 10 min at room temperature, followed by relative luminescent unit (RLU) measurement with a Synergy Multi-Mode microplate reader (BioTek Instruments, Inc., Winooski, VT, USA) at 10 s intervals for 2 h. The production of endogenous ROS was recorded in relative luminescence units and normalised to 100 μg protein of the samples. Hydrogen peroxide (100 μM) and DHCFDA were used as positive controls.

### 4.5. Annexin-V Labelling and Live/Dead Cell Sorting

Approximately 2 × 10^6^ cells were collected, washed with 1× PBS, and centrifuged at 100× *g* for 3 min at 4 °C. All subsequent steps were performed at 4 °C. Cells were labelled with FITC-conjugated Annexin-V (Biolegend, San Diego, CA, USA) for 15 min in the dark, centrifuged, and resuspended in 1× PBS. The cells were analysed and sorted on a BD FACSAria™ III flow cytometer (B.D. Biosciences, San Jose, CA, USA). Apoptotic features were evaluated by the size and granularity of the NB4 cells, followed by gating out the FITC-positive cell population. A minimum of 1 × 10^5^ cells were sorted out for further experiments.

### 4.6. Fluorometric Measurement of Calpain Activity

A minimum of 1 × 10^5^ cells were sorted out by flow cytometry based on the FITC-Annexin-V positivity. Then, following the manufacturer’s protocol, cells were harvested and measured with a Calpain Activity Fluorometric Assay Kit (Sigma Aldrich, MAK228-1KT).

### 4.7. Gene Expression Analyses Using Real-Time qPCR

RT-qPCR was conducted with TaqMan probes (ABI, Applied Biosystems, Waltham, MA, USA) for NRF2, GST-Ω, PRDX5, TG2, and GAPDH on a Roche Light Cycler^®^ 480 II (Roche Molecular Systems, Inc., Pleasanton, CA, USA).

### 4.8. SDS-PAGE and Western Blotting

A total of 1–2 × 10^6^ cells were collected and lysed in lysis buffer (50 mM Tris, 1 mM EDTA, MEA, 0.5% Triton X-100, 1 mM PMSF) containing a protease inhibitor cocktail (PIC) (Sigma-Aldrich) at a 1:100 dilution ratio, homogenised with 5–7 strokes of a sonicator at 40% cycle intensity (Branson Sonifer^®^ 450), and centrifuged at a maximum of 13,700× *g* at 4 °C for 15 min. Protein concentrations were measured in triplicate in 96-well flat-bottom plates at 595 nm with a Synergy Multi-Mode Microplate Reader (BioTek Instruments, Inc.) and the Bradford assay (Bio-Rad). Samples were diluted to 2 mg/mL, mixed with equal volumes of 2× SDS denaturation buffer (0.125 M Tris-HCl, pH 6.8, 4% SDS, 20% glycerol, 10% MEA, and 0.02% bromophenol blue), and incubated at 99 °C for 10 min. Proteins were separated on 8–10% SDS-polyacrylamide gels, blotted onto a PVDF membrane (Millipore, Burlington, MA, USA), blocked with 5% non-fat dry milk in 1× Tris-buffered saline containing Tween 20 (TTBS) for 1 h at room temperature, and incubated overnight at 4 °C with primary antibodies diluted in 0.5% milk in TTBS (1:1000–1:5000). After three 15 min washes with TTBS, membranes were incubated for 1 h with horseradish peroxidase-labelled affinity-purified secondary antibody (Advansta, San Jose, CA, USA; 1:10,000–1:20,000). The protein bands were visualised with an ECL kit (Advansta) and quantified using ImageJ software version 1.09.

### 4.9. Statistical Analysis

Statistical analyses were carried out using GraphPad Prism version 9.0.1. with a two-way ANOVA (Bonferroni post hoc test; * *p* < 0.05, ** *p* < 0.01, and *** *p* < 0.001).

## Figures and Tables

**Figure 1 ijms-24-10938-f001:**
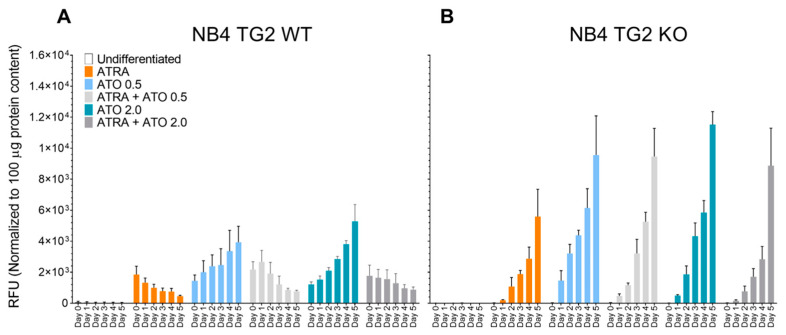
Tissue transglutaminase impairs the capability of ATRA + ATO-differentiated NB4 cells for endogenous ROS production. The production of endogenous ROS was determined for each cell line using a DCFDA-based method and reported in Relative Fluorescence Units (RFU). The graphs represent mean ± SD values normalised to 100 µg protein of total cell lysate content. (**A**) NB4 TG2 WT cells were treated with ATRA 1 µM, ATO 0.5 µM, ATRA 1 µM + ATO 0.5 µM, ATO 2.0 µM, and ATRA 1 µM + ATO 2.0 µM for five days. Endogenous ROS values were measured on each day in triplicate (n = 5). (**B**) NB4 TG2 KO cells were treated with ATRA 1 µM, ATO 0.5 µM, ATRA 1 µM + ATO 0.5 µM, ATO 2.0 µM, and ATRA 1 µM + ATO 2.0 µM for five days. Endogenous ROS values were measured on each day in triplicate (n = 5).

**Figure 2 ijms-24-10938-f002:**
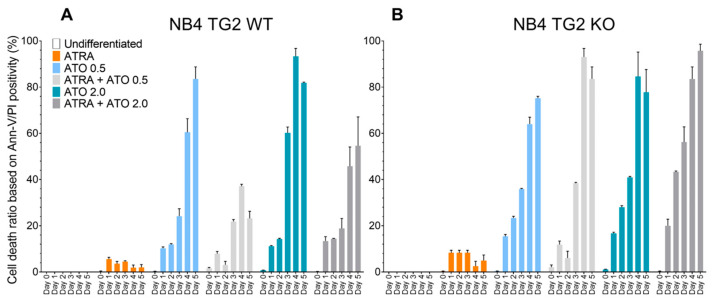
Transglutaminase 2 attenuates ATO-induced cell death. NB4 cells were treated, harvested, and labelled with FITC/PE-conjugated Annexin-V/PI for 15 min. Cells were analysed on a BD FACSAria™ III. Apoptotic features were evaluated by the size and granularity of the NB4 cells, followed by gating out the FITC/PI-positive cell population. (**A**) NB4 TG2 WT cells were treated with ATRA 1 µM, ATO 0.5 µM, ATRA 1 µM+ATO 0.5 µM, ATO 2.0 µM, and ATRA 1 µM + ATO 2.0 µM for five days. Dead cell ratios were measured each day in triplicate (n = 5). (**B**) NB4 TG2 KO cells were treated with ATRA 1 µM, ATO 0.5 µM, ATRA 1 µM + ATO 0.5 µM, ATO 2.0 µM, and ATRA 1 µM + ATO 2.0 µM for five days. Dead cell ratios were measured each day in triplicate (n = 5).

**Figure 3 ijms-24-10938-f003:**
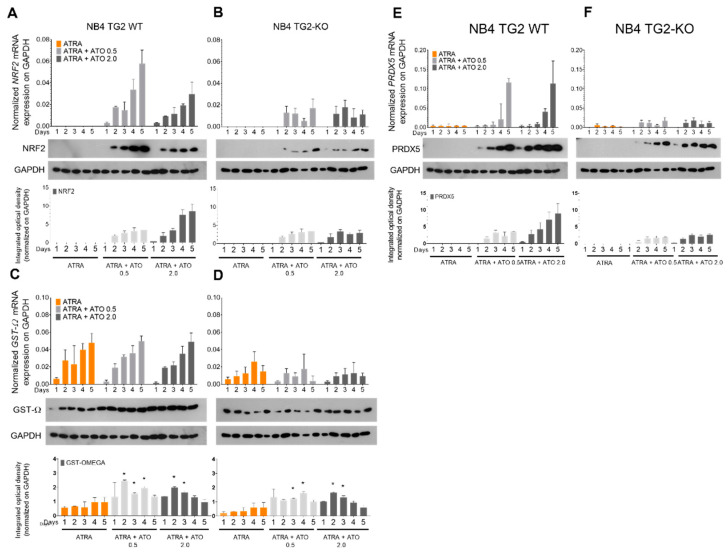
Endogenous ROS production is regulated via the transcription factor NRF2, influencing the expression of antioxidant genes. NB4 TG2 WT and TG2 KO cells were incubated with ATRA 1 µM, ATO 0.5 µM, ATRA 1 µM + ATO 0.5 µM, ATO 2.0 µM, and ATRA 1 µM + ATO 2.0 µM for five days. Relative mRNA expressions of (**A**,**B**) NRF2, (**C**,**D**) GST-Ω, and (**E**,**F**) PRDX5 were measured on the indicated days using real-time Q-PCR and were normalised to GAPDH (means ± SD, n = 3). Representative Western blots/densitometry analysis showed protein expression levels upon various treatments over 5 days (n = 3). Statistical significance was determined via two-way analysis of variance (ANOVA; Bonferroni post hoc test; ATRA vs. ATRA + ATO * *p* < 0.05).

**Figure 4 ijms-24-10938-f004:**
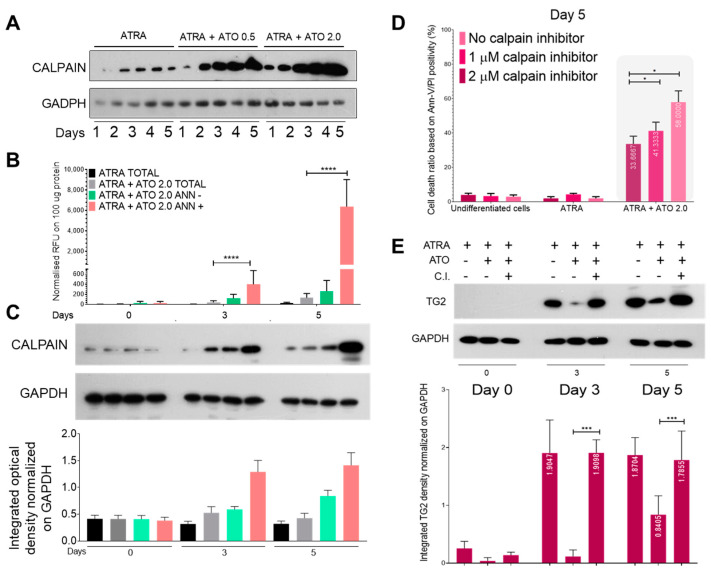
ATO-induced ROS production elevates calpain expression/activation, resulting in increased proteolysis of transglutaminase 2. NB4 WT cells were incubated with ATRA 1 µM, ATRA 1 µM + ATO 0.5 µM, and ATRA 1 µM + ATO 2.0 µM for five days. (**A**) Representative Western blots/densitometry analysis showing calpain total protein expression levels upon treatment over 5 days (n = 3). (**B**) The graph represents relative normalised calpain activity RFU values on 100 µg protein content of the total, Annexin-V positive/negative sorted cell lysates. Samples’ activity values were determined with a Synergy Multireader fluorometer (400/505 nm). (**C**) Representative Western blots/densitometry analysis showing calpain protein expression levels in total and Annexin-V sorted cell lysates (n = 3). (**D**) Untreated, ATRA differentiated, and ATRA + ATO 2.0 μM treated NB4 WT cells were treated with 1 μM or 2 μM calpain inhibitor (C.I.) for 5 days. Flow cytometric analysis shows the cell viability changes upon the treatments. (**E**) Representative Western blots/densitometry analysis showing TG2 protein expression levels in total cell lysates upon calpain inhibitory treatments (n = 3). Statistical significance was determined via two-way analysis of variance (ANOVA; Bonferroni post hoc test; * *p* < 0.05, and *** *p* < 0.001, **** *p* < 0.0001).

**Figure 5 ijms-24-10938-f005:**
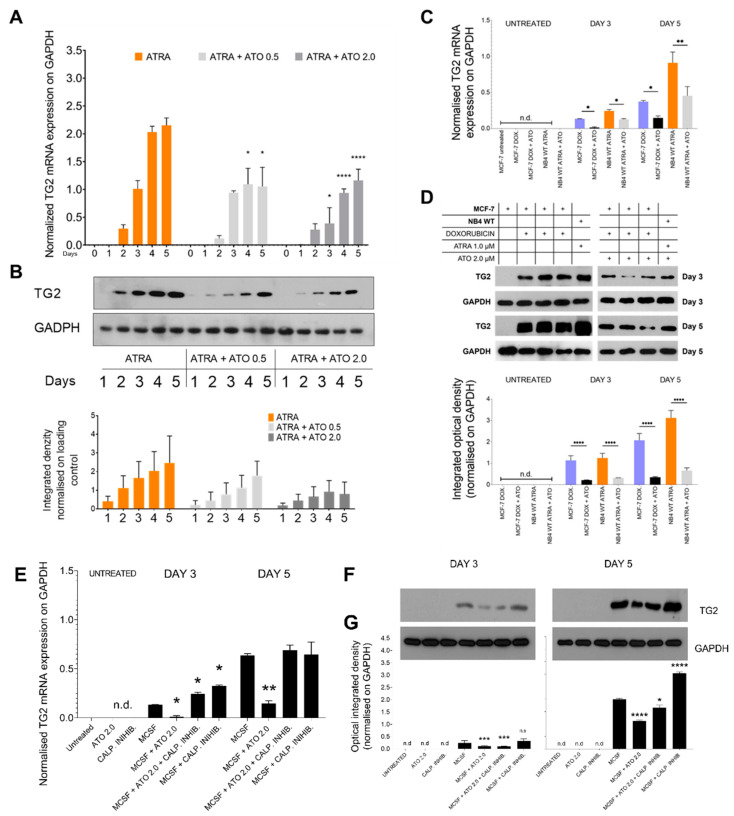
Transglutaminase 2 levels are reduced by ATO treatment. (**A**) Relative mRNA expressions of TG2 were measured in NB4 TG2 cells on the indicated days using real-time Q-PCR and normalised to GAPDH. (**B**) Representative Western blots/densitometry analysis showing TG2 protein expression levels in total cell lysates upon ATRA and ATRA + ATO treatments (n = 3). (**C**) MCF-7 cells were treated with 1 µg/mL doxorubicin (DOX), ATO 2.0 µM, or with a combination. As a positive control, ATRA-treated NB4 WT cells were used. Relative mRNA expressions of TG2 were measured using real-time Q-PCR and normalised to GAPDH. (**D**) Representative Western blots/densitometry analysis showing TG2 protein expression levels in total MCF-7 and NB4 WT cell lysates (n = 3). (E) Relative mRNA expressions of TG2 were measured in monocytes on the indicated days using real-time Q-PCR and normalised to GAPDH. (F) Representative Western blots and (**G**) densitometry analysis showing TG2 protein expression levels in total cell lysates upon ATO 2.0 µM, calpain inhibitor 2.0 µM, MCSF, MSCF + ATO 2.0 µM, MSCF + ATO 2.0 µM + calpain inhibitor 2.0 µM, and MSCF + calpain inhibitor 2.0 µM treatments for 3 and 5 days (n = 3). Statistical significance was determined via two-way analysis of variance (ANOVA; Bonferroni post hoc test; MCSF vs. combination treatments; * *p* < 0.05, ** *p*< 0.01 and *** *p* < 0.001, **** *p* < 0.0001). n.d.—no data.

## Data Availability

The data presented in this study are available on request from the corresponding author.

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
