# Peer review of "ATO Increases ROS Production and Apoptosis of Cells by Enhancing Calpain-Mediated Degradation of the Cancer Survival Protein TG2"

_ijms, 2023, doi:10.3390/ijms241310938_

Round 1
Reviewer 1 Report
Journal of International Journal of Molecular Sciences
Research Article;
The article entitled “ATO increases ROS production and apoptosis of cells by enhancing calpain-mediated degradation of the cancer survival protein TG2’’. The author investigated the arsenic trioxide (ATO) effect on reactive oxygen species. As transglutaminase 2 (TG2) is a critical cancer cell survival factor that activates several signaling pathways to foster drug resistance, cancer stem cell survival, metastasis, inflammation, epithelial mesenchymal transition, and angiogenesis. The author describe the Arsenic trioxide (ATO), alone or in combination with ATRA, can improve outcomes. ATO exerts cytotoxic effects in a variety of ways by inducing oxidative stress, genotoxicity, altered signal transduction, and/or epigenetic modification. The author analyzed that ATO increased ROS production and apoptosis ratios in ATRA-differentiated NB4 leukemia cells and that these responses were enhanced when TG2 was deleted. The combined ATRA+ATO treatment also increased the amount of nuclear factor erythroid 2-related factor 2 (NRF2) transcription factor, an adaptive regulator of cellular oxidative stress response, and proteolytic activity of calpain, resulting in TG2 degradation and reduced survival of WT leukemia cells.
I carefully read the manuscript and it needs minor revision for publication in the journal. There are some common mistakes, references and English language problem in the article which should be corrected by the authors. After the correction of all the mistakes, the article could be considered for publication in the prestigious International Journal of Molecular Sciences Journal.
Comments for Authors
Ø Write at least five keywords and write it in alphabetical order.
Ø The author needs to describe introduction section and include latest references.
Ø Mentioned the original full image size in all Figures.
Ø The author needs to revise the figures legends and put accurate significance rate I-e in Figure 3. The author mentioned complete significance rate but in figure there is only * p < 0.05.
Ø The author needs to include the xenograft tumor model to confirm the effect of ATO.
Ø There are many research publication about effect of ATO on various cancer cells and an induce ROS induce apoptosis. What is the novelty of the study?
Ø Check grammatically and spelling throughout the manuscript. There are many mistakes which needs to revise.
Cite the following references;
v DOI: 10.2174/1871520622666220831124321
v DOI: 10.1038/s41419-021-03771-z

Reviewer 2 Report
The authors demonstrated that the combination of ATRA and ATO, but not ATRA treatment alone, could increase the amount and proteolytic activity of calpain, thereby driving both the degradation of TG2 and the decreased survival of leukemic cells. If the authors can perform any rescue experiment with TG2, the manuscript could be more reasonable and helpful for many readers. It should be considered before publication.
Reviewer 3 Report
This study suggested a promising mechanism by which ATO promotes APL cell death. However, the results are preliminary and additional experiments are required for publication.
Major points:
1. Figure 1 and Figure 2 suggest TG2 attenuates ATO-induced cell death by reducing ROS level. But no clear evidence shows how ROS contribute to cell death here. For example, ROS generation in both ATRA and ATRA+ATO 0.5 treated cells decrease over time, while only ATRA+ATO 0.5 but not ATRA causes elevated cell death. Similarly, ATRA treated TG2 KO cells produce increasing ROS level over time but keep a low cell death rate. Apparently, ROS production may not be a link between TG2 and cell death. Authors should also include an anti-oxidant control to see if inhibition of ROS can rescue cell death.
2. It is not clear whether authors are referring to apoptosis solely when describing “ATO-induced cell death”. A flow cytometry graph should be included to demonstrate how authors define apoptotic cell group. For example, are those dead cells in Figure 2 annexin V+/PI+ or annexin V+/PI-, or both? Did authors check other forms of cell death than apoptosis?
3. Figure 3 shows expression of anti-oxidant genes but cannot draw the conclusion that endogenous ROS is regulated via NRF2.
4. Figure 4 & 5 show almost the same set of results (TG2 mRNA and protein level). However, they support different conclusions. Does ATO treatment down-regulate TG2 at transcription or degradation level, or it affects differentially in various cell types? Authors should provide further evidence whether TG2 is degraded or transcriptionally/translationally downregulated. As TG2 is a reported calpain substrate (PMID: 9648871), authors may want to check the presence of cleaved TG2. Authors may also use proteasome inhibitor to see if TG2 level (as well as cell death phenotype) is rescued.
5. In Figure 4B, both ATRA+ATO ANN- and ATRA+ATO ANN+ show higher calpain activity than ATRA+ATO TOTAL, which would be a concern, as ATRA+ATO TOTAL is supposed to be a weighted average of ATRA+ATO ANN- and ATRA+ATO ANN+ and valued between them.
6. In the method section, authors describe the way to isolation human monocytes. Ficoll layer containing monocytes was washed in saline, incubated with CD14+ microbeads, and centrifuged at 700g. That seemingly does not separate CD14+ cells because all cells including microbeads would be in the pellet. Could authors elaborate on this?
Minor points:
1. Line 90 writes “similar endogenous ROS production patterns were seen in NB4 TG2-KO”. Obviously, they are not similar.
2. As the lysis buffer contains a high concentration of Triton X-100, Bradford assay may not be appropriate as per Bio-Rad guidance.
3. X-axis labels in most figures are too small to read. Please enhance the resolution if possible.
English styles are good except some typos.
Round 2
Reviewer 3 Report
The authors have made great improvement and all my points are well addressed. I only have a few further comments.
In regard to major point 3, authors are trying to attribute the reduced ROS level in NB4 TG2 WT cells to elevated NRF2 and antioxidant genes, if I'm correct. However, the subtitle in lines 130~131 is ambiguous. Does ROS production or NRF2 regulates expression of antioxidant genes? Does NRF2 promotes or reduced ROS production? Authors could be more precise on this subtitle. For example, "ROS level may be regulated by NRF2 signaling" or "reduced ROS level is associated with enhanced NRF2 signaling", etc.
In regard to minor point 1, now I understand that the similarity refers to ATO alone. It's better to explicitly state this in the main text.
Author Response
Based on the comment of the reviewer the manuscript results section about the ros production has been changed to the following:
"ROS-induced nuclear factor Erythroid 2-related factor 2 (NRF2) transcription factor mediates the expression of antioxidant genes in a TG2 dependent manner"
Regarding the minor point, the ROS production-related sentence also has been modified as the following:
"Similar endogenous ROS production patterns were seen in NB4 TG2-KO cells upon ATO treatment alone"